# The Impact of Territorial Spatial Transformation on Carbon Storage: A Case Study of Suqian, East China

**Wenting Huang [1], Long Guo [2], Ting Zhang [1,*], Ting Chen [2], Longqian Chen [1], Long Li [1,3]** and **Xundi Zhang [1]**

1   School of Public Policy and Management, China University of Mining and Technology, Daxue Road 1, Xuzhou 221116, China; wt_huang@cumt.edu.cn (W.H.); chenlq@cumt.edu.cn (L.C.); long.li@cumt.edu.cn (L.L.); ts21090072a31ld@cumt.edu.cn (X.Z.)

2   Land Expropriation and Survey Center of Suqian City, Hongzehu Road 793, Suqian 223800, China; 18860800099@163.com (L.G.); ctlow@126.com (T.C.)

3   Department of Geography, Earth System Sciences, Vrije Universiteit Brussel, Pleinlaan 2, 1050 Brussels, Belgium

*   Correspondence: tingzhang@cumt.edu.cn; Tel.: +86-516-8359-1327

**Abstract:** The carbon storage of terrestrial ecosystems plays a crucial role in mitigating climate change, and the transformation of territorial space has a significant impact on the carbon cycle of a country's terrestrial ecosystems. Therefore, evaluating the impact of space transformation on carbon storage is essential for enhancing regional carbon storage potential and reducing carbon emissions. We use the Integrated Valuation of Ecosystem Services and Tradeoffs (InVEST) model to analyze the dynamic changes in territorial spatial transformation and carbon storage from 2000 to 2020 in Suqian, as well as their relationship. On this basis, the optimization strategy and specific path for improving territorial space carbon storage capacity were determined. The results show the following: that (1) from 2000 to 2020, territorial spatial transformation in Suqian was dramatic, with the most significant changes occurring between 2005 and 2010. The scale of mutual transformation between agricultural production space and urban–rural construction space was the largest. (2) Carbon storage gradually decreased in Suqian City, with a total reduction of $1.23 \times 10^6$ tons over 20 years and an annual decrease of 1.46%. The carbon density of forested space was significantly higher than that of other spaces. The conversion of agricultural production space and forestland space to urban–rural construction space was the main factor driving a decrease in carbon storage. (3) Territorial spatial transformation is a spatial manifestation of the evolution of human–land relationships. Regulating the function, scale, structure and layout of territorial space as a whole and implementing differentiated management of specific space will be beneficial to optimize carbon storage in Suqian.

**Keywords:** territorial spatial transformation; carbon storage; InVEST model; optimization strategy; Suqian City

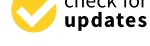



## 1. Introduction

Terrestrial ecosystems, including their carbon reservoirs such as plants and soil, which can store approximately three times the atmospheric amount, are an important part of global carbon stock and play a critical role in mitigating climate change [1,2]. The change in carbon storage capacity in terrestrial ecosystems is often affected by natural disturbances and human activities [3]. However, the changes in land space due to the impact of human activities are greater than those of natural disturbances [4]. According to data from the Intergovernmental Panel on Climate Change (IPCC), land use change currently contributes $1.5 \times 10^9$ tons of carbon emissions per year, making it one of the largest sources of carbon increase in the atmosphere, second only to fossil fuel combustion [5]. Changes in territorial space directly affect the ability of vegetation and soil to sequester carbon, which plays a critical role in carbon storage fluctuations in terrestrial ecosystems, resulting in dynamic changes in carbon storage [6,7]. With the ongoing urbanization, many regions frequently

engage in "random construction and alteration" for short-term economic benefits, resulting in extremely low space use efficiency and the wasteful use of arable land resources and ecological space [8]. Large areas of high-carbon-density land, such as arable land, forestland and grassland, are being transformed into low-carbon-density urban construction land, resulting in a significant reduction in carbon storage in terrestrial ecosystems, seriously threatening the sustainable development of regional ecosystems [9,10]. Therefore, determining how to ensure the carbon storage of urban terrestrial ecosystems, while scientifically planning the development of the territorial space, will be a major challenge facing future urban development in China.

The transformation of the territorial space refers to the application of land transformation theory in the research of new era territorial governance. The essence of territorial spatial transformation is the mapping of the spatial evolution of the human–earth relationship [11]. With the developing society, people's basic needs exist within a relational hierarchy encompassing survival, safety the pursuit of material wealth and satisfying spiritual needs. Therefore, as the process of ecological awareness grows in a civilization, people tend to optimize production space and urban space through technological progress and efficiency gains, to maintain and improve the scale, structure and function of ecological space to support high-quality regional economic and social development. In the relevant theoretical research applying the low-carbon concept in territorial spatial transformation, scholars identify the carbon source/sink function of different types of land use, such as forests, wetlands and industrial land [12–14]. Earlier research has been carried out on the carbon balance of land use and the optimization of low-carbon territorial space, the results of which have been widely used in practice, but it has mainly focused on the scale and structure of land use. Further research is needed on the transformation of territorial space patterns and how they affect carbon storage.

The rapid urbanization and industrialization in China have caused frequent conversion of different landscapes and increasing conflicts, which greatly reduce the carbon sequestration capacity of ecosystems [15]. Against the backdrop of global climate change, the Chinese government has made a commitment to achieve a carbon peak by 2030 and carbon neutrality by 2060. Enhancing the carbon storage capacity of terrestrial ecosystems is one of the important paths to achieve carbon neutrality, which requires optimizing the configuration and coordinated development of different territorial spaces to support these dual carbon goals [16]. The ecological impact caused by the evolution of land spatial patterns has regional and aggregational characteristics, and the impact on regional ecosystem carbon storage varies greatly over space and time [17,18], resulting in uncertainty in the spatiotemporal distribution patterns and variation mechanisms of carbon sources/sinks in terrestrial ecosystems. Therefore, quantifying the relationship between territorial spatial transformation and carbon storage is conducive to the formulation of space use strategies, under the goal of enhancing carbon storage capacity and achieving the goal of regulating climate change. Previous studies on the impact of land use or space changes on ecosystem carbon storage have provided important insights and directions for this paper. From the perspective of carbon storage estimation methods, the InVEST model has fast information processing, low data demand and high accuracy. It can visualize the changes and spatial distribution of carbon storage and has been widely used in land use-related carbon storage estimation [19,20]. For example, Leh et al. used the InVEST model to analyze the impact of land use and land cover changes on carbon storage from 2000 to 2009 [21]. Nelson et al. also used InVEST to evaluate the impact of global urban land changes on carbon storage [22]. With further research, many scholars have combined land-use simulation models such as SLEUTH and FLUS with the InVEST model to explore the changes in carbon storage under different scenarios in the future [23,24]. Zhu et al. studied the spatial distribution of carbon storage in the Qi River Basin under different scenarios and explored the impact of land use change on the spatial distribution of carbon storage under different scenarios [25]. Guo et al. simulated the land use cover and carbon storage distribution of the Taihang Mountains in 2035 under three different development scenarios based on the patch-generating land use

simulation (PLUS) model [26]. Zhao et al. used PLUS-InVEST and gravity center model to analyze the spatial-temporal evolution of carbon storage in different scenarios, and used geodetector to clarify the driving mechanism of carbon storage [27]. However, most previous empirical and modeling studies have focused on the impact of land use/cover change(LULC)changes on carbon storage, and there are few studies starting from the perspective of land use governance, combining spatial planning policies and studying the transformation of "space use" and its carbon storage effect.

Suqian is a typical resource–industry composite city located in the northern part of Jiangsu Province, China. It boasts superior natural resources and developed water systems and is the only prefecture-level city in the country with a specific geographical indication of "two lakes and two rivers". It is known as the "green heart of Jiangsu and the green lung of East China". The implementation of the joint development strategy in the Jiangsu region has increased government support for the northern Jiangsu region, and the future will be a golden period of rapid industrialization and urbanization and comprehensive economic and social progress for Suqian. It can be foreseen that the demand for urban–industrial–mining space in Suqian will remain high for a long time, and the space for arable land and ecological land will be further reduced, which will greatly change the regional space use pattern and ecosystem pattern and lead to a decline in ecosystem services (such as carbon storage). Therefore, taking Suqian City as an example, this study explores the territorial spatial transformation of typical urban environments and their carbon storage effects, combined with territorial space governance policies. This has important theoretical and practical significance for enriching and improving the theory of territorial spatial transformation, promoting regional carbon neutrality with space use optimization, and promoting sustainable development of the ecological environment in the research area. The specific objectives of this study are as follows: (1) With the help of analysis methods such as transfer matrix and transfer map, summarize the territorial spatial transformation characteristics of Suqian from 2000 to 2020. (2) Using the InVEST model to estimate carbon storage, divide carbon source and sink areas, and determine the impact of different territorial spatial transformation on carbon storage. (3) Analyze the impact mechanism of territorial space transformation on carbon storage changes and propose strategies for optimizing carbon storage capacity based on territorial space governance policies.

## 2. Materials and Methods

### 2.1. Study Area

The research area is Suqian City, which is known as the central city of the east terminal of the New Eurasian Land Bridge urban agglomeration. It is located at a latitude of 33°8′–34°25′ north and a longitude of 117°56′–119°10′ east, and it belongs to Jiangsu Province, which has efficient transportation infrastructure and a superior natural environment and economic conditions, covering an area of 8522 km$^2$ and governing two districts and three counties (Sucheng District, Suyu District, Sihong County, Sihong River County and Shuyang County) (Figure 1). By the end of 2020, the city's permanent resident population reached 4.9862 million. The area is located in the warm temperate zone with a monsoon climate, characterized mainly by plains landforms. As a result of increased human activity related to population growth and industrial activities, the spatial utilization intensity in Suqian City has been increasing. The recent changes in this city can be seen as a microcosm of China's transition and development toward resource-oriented urban spaces.

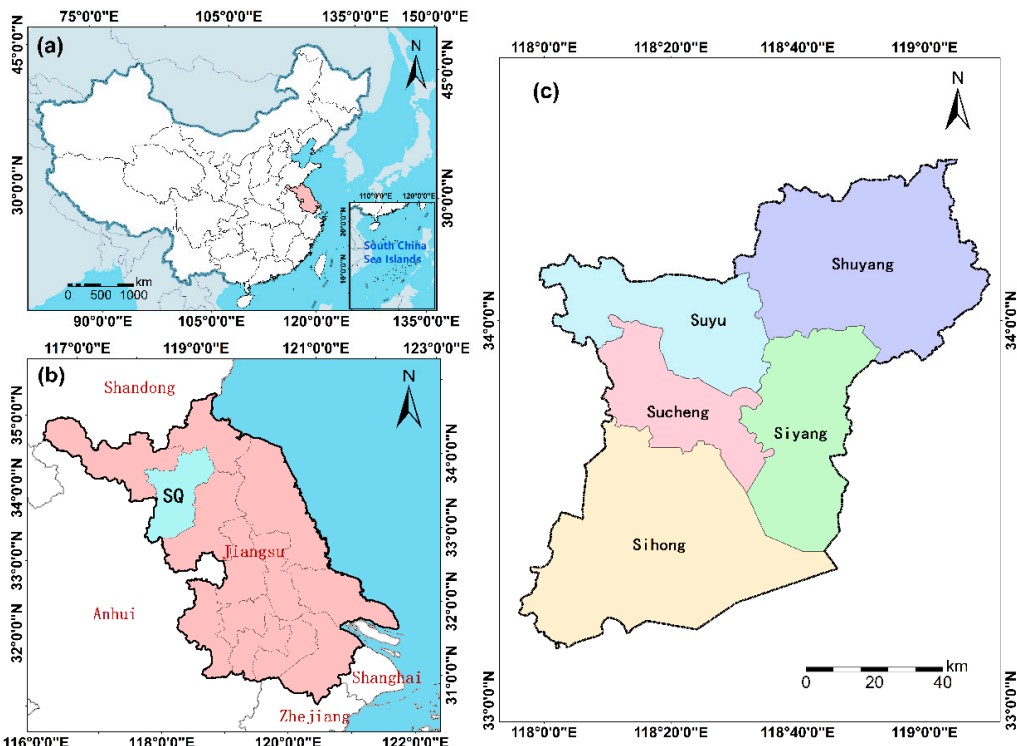

**Figure 1.** Geographical location of the study area. (**a**) Location of Jiangsu in China. (**b**) Location of Suqian in Jiangsu. (**c**) Boundary and administrative divisions of Suqian.

*2.2. Data Source*

This study utilizes digital land use data for territorial space classification. We used 30 m spatial resolution land use data from 2000, 2005, 2010, 2015 and 2020, with all data geographically projected onto the WGS-84 coordinate system using ArcGIS. The data were obtained from the Land Use Status Database of the Chinese Academy of Sciences Resource and Environmental Science Data Center (https://www.resdc.cn/, accessed on 20 March 2022), and the detailed process for extracting this dataset can be found in the relevant literature [28]. This database has two main advantages. The first is its high-resolution data and continuous time coverage spanning five periods. The second feature is the high accuracy of the dataset, which meets the requirements of this study. Based on the second-level classification standard in the National Remote Sensing Monitoring Land Use/Cover Classification System, the corresponding relationship between territorial space types and land use types has been determined. The territorial space in Suqian City was classified into six types, namely agricultural production space (agricultural space), including paddy fields and drylands; urban–rural construction space (urban–rural space), including urban land, industrial land, transportation land and rural residential areas; forestland space, including forestland, shrub land, sparse forestland and other forestland; grassland space, including high-coverage grassland, medium-coverage grassland and low-coverage grassland; wetland space, including rivers, lakes, reservoirs, ponds and sandbars; and other space, including marshland, bare land and rocky terrain. Temperature and precipitation data were obtained from the National Meteorological Data Center (https://data.cma.cn/, accessed on 15 May 2022).

*2.3. Methods*

2.3.1. Dynamic Degree of Territorial Space

The dynamic degree can explain the speed of regional territorial spatial transformation, including the dynamic degree of a single type and the comprehensive dynamic degree [29].

$$D = \frac{U_b - U_a}{U_a} \times \frac{1}{T} \times 100\% \tag{1}$$

In Formula (1), $D$ represents the dynamic degree of a certain type of territorial space; $U_a$ is the area of a certain type of "territorial space category" at the beginning; $U_b$ is the area of a certain type of "territorial space category" at the end; and $T$ represents the research time.

$$LC = \left[ \frac{\sum_{i=1}^{n} \Delta LU_{i-j}}{2 \cdot \sum_{i=1}^{n} \Delta LU_i} \right] \times \frac{1}{T} \times 100\%$$

(2)

In Formula (2), $LC$ represents the comprehensive dynamic degree of "national land spatial categories"; $LU$ is the area of the $i$-th type of "territorial space category" at the beginning of the study period; $\Delta LU_{i-j}$ represents the area that has been transformed from type $i$ to non-type $i$ "national land spatial categories" during the research period; and n is the total number of regional "national land spatial categories".

### 2.3.2. Transition Matrix of Territorial Space

The territorial space transition matrix can not only explain the structure at a certain point in time, but also quantitatively describe the dynamic process of mutual transformation of various types of territorial space.

$$S_{ij} = \begin{bmatrix} S_{11} & S_{12} & S_{13} & \cdots & S_{1n} \\ S_{21} & S_{22} & S_{23} & \cdots & S_{2n} \\ S_{31} & S_{32} & S_{32} & \cdots & S_{3n} \\ \vdots & \vdots & \vdots & \vdots & \vdots \\ S_{n1} & S_{n2} & S_{n3} & \cdots & S_{nn} \end{bmatrix}$$

(3)

In Formula (3), $S$ represents the territorial space area and $S_{ij}$ represents the area transformed from the $i$-th type of territorial space to the $j$-th type of territorial space. Each row represents the number of conversions from $i$-th space to other spaces, and each column represents the number of conversions from other spaces to $j$-th space.

### 2.3.3. Contribution Rates of Carbon Storage in the Territorial Space

This study drew on the calculation method of regional ecosystem service value for land use change and established a carbon storage contribution rate index, which represents the proportion of the carbon storage change caused by each type of territorial spatial transformation to the overall carbon storage change in the period [30].

$$R_C = \frac{(C_{t2} - C_{t1}) \cdot S_i}{\Delta C}$$

(4)

In Formula(4), $Rc$ represents the carbon storage contribution rate; $C_{t2}$ and $C_{t1}$ represent the comprehensive carbon density corresponding to the end and beginning of the research period, respectively, reflected by the transformation of territorial space type $i$; $S_i$ represents the area of this type of space change; and $\Delta C$ represents the change in total carbon storage during the study period.

### 2.3.4. Integrated Valuation of Ecosystem Services and Tradeoffs (InVEST) Module and Carbon Density

The carbon module [31] in InVEST 3.10.2 was used to analyze the changes in carbon storage of the territorial space in Suqian City. The carbon module calculates the carbon storage of ecosystems through carbon density data and land use data. It divides the carbon storage of ecosystems into the following four carbon pools: aboveground carbon pool (surface vegetation), underground carbon pool (plant roots), soil carbon pool (organic carbon in soil) and dead organic matter carbon pool (dead vegetation and woody debris). According to the prompts in the InVEST model user manual, the necessary basic data for model operation include the terrestrial spatial raster map and the four basic carbon

storage carbon density tables, corresponding to each space type in the map. The calculation formula of this model is as follows:

$$C_i = C_{i\text{-}above} + C_{i\text{-}below} + C_{i\text{-}dead} + C_{i\text{-}soil} \tag{5}$$

$$C_{i\text{-}totle} = C_i \times S_i \tag{6}$$

In the formula, $i$ represents a certain type of territorial space; $C_i$ represents the total carbon density of type $i$, $C_{i\text{-}above}$ represents the aboveground vegetation carbon density of territorial space type $i$, $C_{i\text{-}below}$ represents the underground active root carbon density of territorial space type $i$, $C_{i\text{-}soil}$ represents the carbon density of soil territorial space type $i$ and $C_{i\text{-}dead}$ represents the carbon density of vegetation litter of territorial space type $i$, with units of t/hm$^2$. $C_{i\text{-}total}$ represents the total carbon storage, measured in t, and $S_i$ represents the total area of territorial space type $i$, measured in km$^2$.

The carbon density data for different territorial space types were constructed based on existing research (Table 1). As prioritizing the use of carbon density results from within Jiangsu province will help improve the assessment accuracy [32], vegetation and soil type carbon density at regional or national scales were used as a complement [33–36]. Next, based on studies by Alam and Giardina [37,38], carbon density was corrected, in combination with temperature and precipitation factors. The annual average temperature and precipitation data of Suqian and Jiangsu province were taken into the following formulas, respectively, and the ratio of the two was the correction coefficient. The product of the carbon density data and the correction coefficient of Jiangsu province was the Suqian carbon density data.

$$C_{SP} = 3.3968 \times MAP + 3.9961 \tag{7}$$

$$C_{BP} = 6.798 \times e^{0.0054 \times MAP} \tag{8}$$

$$C_{BT} = 28 \times MAT + 398 \tag{9}$$

$$K_B = K_{BP} \times K_{BT} \,, \; K_{BP} = \frac{C'_{BP}}{C''_{BP}}, \; K_{BT} = \frac{C'_{BT}}{C''_{BT}} \tag{10}$$

$$K_S = \frac{C'_{SP}}{C''_{SP}} \tag{11}$$

**Table 1.** Carbon density of each territorial space in Suqian (t/hm$^2$).

| Territorial Space | Carbon Density (t/hm$^2$) | | | |
|---|---|---|---|---|
| | **Above** | **Below** | **Soil** | **Dead** |
| Agricultural space | 5.387 | 1.024 | 91.886 | 1.000 |
| Forestland space | 18.909 | 7.564 | 125.910 | 3.800 |
| Grassland space | 2.056 | 10.031 | 98.612 | 0.190 |
| Wetland space | 1.023 | 0.019 | 72.203 | 0.010 |
| Urban–rural space | 0.575 | 0.117 | 80.215 | 1.200 |
| Other space | 0.127 | 0.064 | 73.786 | 0.010 |

In these formulas, $C_{SP}$ represents the soil carbon density derived from annual precipitation and $C_{BP}$ and $C_{BT}$ represent the biomass carbon density derived from annual precipitation and annual average temperature, respectively, in units of t/hm$^2$. $MAP$ represents the annual mean precipitation in mm. $MAT$ is the average annual temperature in °C. $K_B$ represents biomass carbon density correction coefficient. $K_{BP}$ represents the biomass carbon density correction coefficient of the precipitation factor, $K_{BT}$ represents the biomass carbon density correction coefficient of the temperature factor, and $K_S$ represents the soil carbon density correction coefficient. $C'$ and $C''$ represent carbon density data for Suqian City and Jiangsu Province.

## 3. Results

### 3.1. Analysis of the Spatiotemporal Evolution of Territorial Space Use in Suqian City

3.1.1. The Transformation Intensity of Territorial Space

The transformation intensity of the "territorial space" in Suqian City was measured according to its degree of dynamic change. The first concept measured was the intensity of each type (Table 2). From the absolute values of the dynamic degree of various "territorial space" types, the changes in the "territorial space" in Suqian City from 2000 to 2020 were presented as follows: grassland space > urban–rural space > forestland space > agricultural space > wetland space. Except for the maximum values of the dynamic degrees of grassland and other ecological spaces that appeared in 2015–2020 and 2010–2015, respectively, the maximum values of other cover types were reached during 2005–2010. Secondly, the intensity of each period was measured. The comprehensive dynamic degree of the "territorial space" in Suqian City over the 20 years study period was 0.45%, among which the value during 2015–2020 was the highest, and the value during 2000–2005 was the lowest. This indicates that the "territorial space" transformation in Suqian City has been the most intense over the past five years. The third measurement was the intensity of each region. From the comprehensive dynamics of each prefecture-level administrative region in the past 20 years, Sucheng District experienced the greatest change characteristics, followed by Siyang County, Suyu District, Shuyang County and Sihong County, all of which exceeded 0.25%. During 2015–2020, the comprehensive dynamic change characteristics of "territorial space" in each county exceeded 0.6%, and Sucheng District and Suyu District exceeded 1% during 2005–2010, indicating that their "territorial space" transformation was more intense during these periods (Figure 2).

**Table 2.** Dynamic degree of each territorial space in Suqian.

| Territorial Space | Dynamic Degree | | | | |
|---|---|---|---|---|---|
| | 2000–2005 | 2005–2010 | 2010–2015 | 2015–2020 | 2000–2020 |
| Agricultural space | −0.08% | −1.18% | −0.13% | −0.32% | −0.42% |
| Urban–rural space | 0.51% | −1.74% | −0.55% | −0.63% | −0.59% |
| Forestland space | −0.03% | −0.02% | 0.06% | 4.17% | 1.32% |
| Grassland space | 0.20% | 0.23% | 0.00% | 0.16% | 0.15% |
| Wetland space | 0.08% | 3.59% | 0.36% | 0.61% | 1.22% |
| Other space | −1.96% | −6.43% | −15.61% | 0.04% | −4.33% |
| Total | 0.06% | 0.80% | 0.10% | 0.82% | 0.45% |

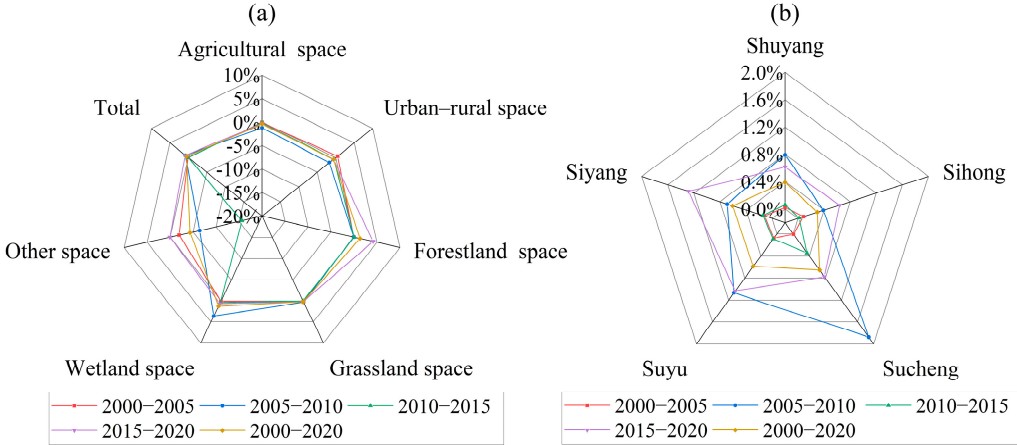

**Figure 2.** (**a**) Dynamic degree of various types of territorial space. (**b**) Dynamic degree of various districts.

3.1.2. The Territorial Space Transition Matrix

The territorial space transition matrix is shown in Tables 3–6. According to the data, significant changes in various space types, including agricultural space, forestland space, grassland space, and urban–rural space, occurred in Suqian City from 2000 to 2020. Among them, the change in agricultural space area was the biggest, with a net decrease of 450.24 km$^2$. Agricultural space was mainly converted into urban–rural space, part of which was turned into wetland space. The proportions of agricultural space converted to urban–rural spaces from 2000 to 2005, 2005 to 2010 and 2010 to 2015 were 80.53%, 90.74% and 95.67%, respectively, but, during 2015–2020, this decreased to 88.28% (Table 7). Among the space types converted from agricultural space, wetland space accounted for 10.54%. In addition, the largest converted area was from urban–rural spaces to agricultural spaces, reaching 284.79 km$^2$, followed by wetland space, with an area of 33.44 km$^2$.

**Table 3.** The territorial space conversion in Suqian City from 2000 to 2005 (km$^2$).

| 2000 | 2005 | | | | | | |
| | Agricultural Space | Forestland | Grassland | Wetland | Urban–rural | Other | Total |
| --- | --- | --- | --- | --- | --- | --- | --- |
| Agricultural space | 5232.89 | 3.19 | 0.19 | 25.52 | 120.43 | 0.22 | 5382.46 |
| Forestland | 1.82 | 52.94 | 0.00 | 0.67 | 0.87 | 0.00 | 56.30 |
| Grassland | 0.22 | 0.00 | 12.90 | 0.91 | 0.16 | 0.00 | 14.19 |
| Wetland | 11.16 | 0.76 | 0.95 | 1353.83 | 2.52 | 0.00 | 1369.24 |
| Urban–rural | 114.20 | 0.84 | 0.13 | 2.01 | 1579.45 | 0.09 | 1696.69 |
| Other space | 0.55 | 0.00 | 0.00 | 0.00 | 0.09 | 2.63 | 3.26 |
| Total | 5360.83 | 57.74 | 14.16 | 1382.93 | 1703.53 | 2.94 | 8522.13 |

**Table 4.** The territorial space conversion in Suqian City from 2005 to 2010 (km$^2$).

| 2005 | 2010 | | | | | | |
| | Agricultural Space | Forestland | Grassland | Wetland | Urban–Rural | Other | Total |
| --- | --- | --- | --- | --- | --- | --- | --- |
| Agricultural space | 4951.10 | 3.84 | 0.11 | 33.95 | 371.81 | 0.03 | 5360.83 |
| Forestland | 4.73 | 47.65 | 0.00 | 0.91 | 4.45 | 0.00 | 57.74 |
| Grassland | 0.12 | 0.00 | 13.41 | 0.53 | 0.11 | 0.00 | 14.16 |
| Wetland | 13.79 | 0.41 | 0.55 | 1362.12 | 6.06 | 0.00 | 1382.93 |
| Urban–rural | 74.48 | 0.80 | 0.08 | 1.54 | 1626.59 | 0.04 | 1703.53 |
| Other | 0.44 | 0.00 | 0.00 | 0.00 | 0.56 | 1.93 | 2.94 |
| Total | 5044.66 | 52.70 | 14.15 | 1399.05 | 2 009.57 | 1.99 | 8522.13 |

**Table 5.** The territorial space conversion in Suqian City from 2010 to 2015 (km$^2$).

| 2010 | 2015 | | | | | | |
| | Agricultural Space | Forestland | Grassland | Wetland | Urban–Rural | Other | Total |
| --- | --- | --- | --- | --- | --- | --- | --- |
| Agricultural space | 4991.34 | 0.23 | 0.01 | 2.08 | 51.00 | 0.00 | 5044.66 |
| Forestland | 0.19 | 50.77 | 0.00 | 0.12 | 1.62 | 0.00 | 52.70 |
| Grassland | 0.05 | 0.00 | 13.94 | 0.10 | 0.06 | 0.00 | 14.15 |
| Wetland | 1.56 | 0.10 | 0.23 | 1396.68 | 0.47 | 0.00 | 1399.05 |
| Urban–rural | 17.72 | 0.16 | 0.01 | 0.37 | 1991.30 | 0.01 | 2009.57 |
| Other | 0.00 | 0.00 | 0.00 | 0.00 | 1.56 | 0.43 | 1.99 |
| Total | 5010.87 | 51.27 | 14.19 | 1399.35 | 2046.02 | 0.44 | 8522.13 |

**Table 6.** The territorial space conversion in Suqian City from 2015 to 2020 (km$^2$).

| 2015 | 2020 | | | | | | |
|---|---|---|---|---|---|---|---|
| | Agricultural Space | Forestland | Grassland | Wetland | Urban–Rural | Other | Total |
| Agricultural space | 4573.42 | 4.25 | 1.82 | 45.20 | 386.18 | 0.00 | 5010.87 |
| Forestland | 4.24 | 40.56 | 0.91 | 2.14 | 3.42 | 0.00 | 51.27 |
| Grassland | 0.15 | 0.00 | 10.04 | 3.84 | 0.15 | 0.00 | 14.19 |
| Wetland | 33.66 | 2.32 | 2.34 | 1351.35 | 9.69 | 0.00 | 1399.35 |
| Urban–rural | 321.36 | 2.57 | 2.81 | 8.41 | 1710.77 | 0.10 | 2046.02 |
| Other | 0.03 | 0.00 | 0.00 | 0.00 | 0.07 | 0.34 | 0.44 |
| Total | 4932.85 | 49.70 | 17.92 | 1410.94 | 2110.27 | 0.44 | 8522.13 |

**Table 7.** Annual change in territorial space use types in Suqian from 2000–2020.

| Type | Amount of Change (km$^2$) | | | | Rate of Change (%) | | | |
|---|---|---|---|---|---|---|---|---|
| | 2000–2005 | 2005–2010 | 2010–2015 | 2015–2020 | 2000–2005 | 2005–2010 | 2010–2015 | 2015–2020 |
| Agricultural space | −21.62 | −316.18 | −33.79 | −78.02 | −0.08 | −1.18 | −0.13 | −0.32 |
| Forestland | 1.43 | −5.03 | −1.44 | −1.56 | 0.51 | −1.74 | −0.55 | −0.63 |
| Grassland | −0.02 | −0.01 | 0.04 | 3.73 | −0.03 | −0.02 | 0.06 | 4.17 |
| Wetland | 13.70 | 16.12 | 0.30 | 11.59 | 0.20 | 0.23 | 0.00 | 0.16 |
| Urban–rural | 6.84 | 306.05 | 36.44 | 64.26 | 0.08 | 3.59 | 0.36 | 0.61 |
| Other | −0.32 | −0.94 | −1.56 | 0.00 | −1.96 | −6.43 | −5.61 | 0.04 |

From 2000 to 2020, urban–rural space was the main hotspot of territorial spatial transformation. Urban–rural space was mainly converted from agricultural space, with a conversion rate of over 95% from 2000 to 2005, 2005 to 2010 and 2010 to 2015. The conversion rate throughout the entire period from 2000 to 2020 was 96.86%, which was much higher than that in other spaces. The expansion of urban–rural space mainly encroached on agricultural space, followed by water bodies and wetlands and forest, with areas of 684.42 km$^2$, 13.57 km$^2$ and 7.11 km$^2$, respectively.

The increase in grassland space was mainly due to the large-scale conversion of agricultural space and wetland space, with 3.37 km$^2$ and 2.12 km$^2$ being converted, respectively. This is closely related to the local urbanization strategy and wetland space construction. The direction of grassland conversion was mainly toward wetlands, with transfer rates of 70.36% and 76.01% for the periods of 2000–2005 and 2005–2010, respectively, which increased to 92.67% during the period of 2015–2020, mainly due to seasonal changes in riparian grasslands.

There was a significant diversification trend in the transfer of forestland space. From 2000 to 2020, the areas of forestland converted to agricultural space and urban–rural were 7.35 km$^2$ and 7.13 km$^2$, respectively. The main sources of the increase in forest area were agricultural spaces, wetlands and urban–rural land, accounting for 58.53%, 20.96% and 20.50%, respectively.

The transfer direction of wetland space in Suqian City has remained stable, mainly being converted to agricultural space. The transfer rates for the periods of 2000–2005, 2005–2010, 2010–2015 and 2015–2020 were 72.51%, 66.23%, 66.03% and 70.11%, respectively. The main source of space transferred into wetland space was agricultural space, which amounted to 81.81 km$^2$. The differences in the areas transferred from forestland, grassland and urban–rural space was not significant.

According to the territorial space status map of Suqian in 2000, 2005, 2010, 2015 and 2020 (Figure 3), the main types of territorial space in Suqian are agricultural and urban–rural spaces. The southwest and northwest parts of Suqian belong to the hilly and ridged

area, where forestland space is widely distributed. Fruit trees are mainly grown in the southwest, while the northwest is rich in forest resources. Agricultural space is mainly distributed in the vast plains, with the northeast being a rice-producing area, the southeast being an area where aquatic products and rice intersect, and the central region being an important producer of flowers and plants. Grassland spaces depend on rivers and lakes and are sparsely distributed in wetland areas in the northwest and southeast, along riverbanks and hills. The urban–rural space is concentrated in the central urban area, and there is an obvious clustered structure in Suyu District and Sucheng District. Residential areas are consistently associated with agricultural spaces and are widely distributed in the central and eastern regions. Wetland space is mainly located in the Jing–Hang Canal—Ancient Yellow River Ecological Zone, which runs through the city's central region, including various water systems such as Luoma Lake and Hongze Lake in the southwest and southeast, as well as the Huaihong Xin River and Liutang River.

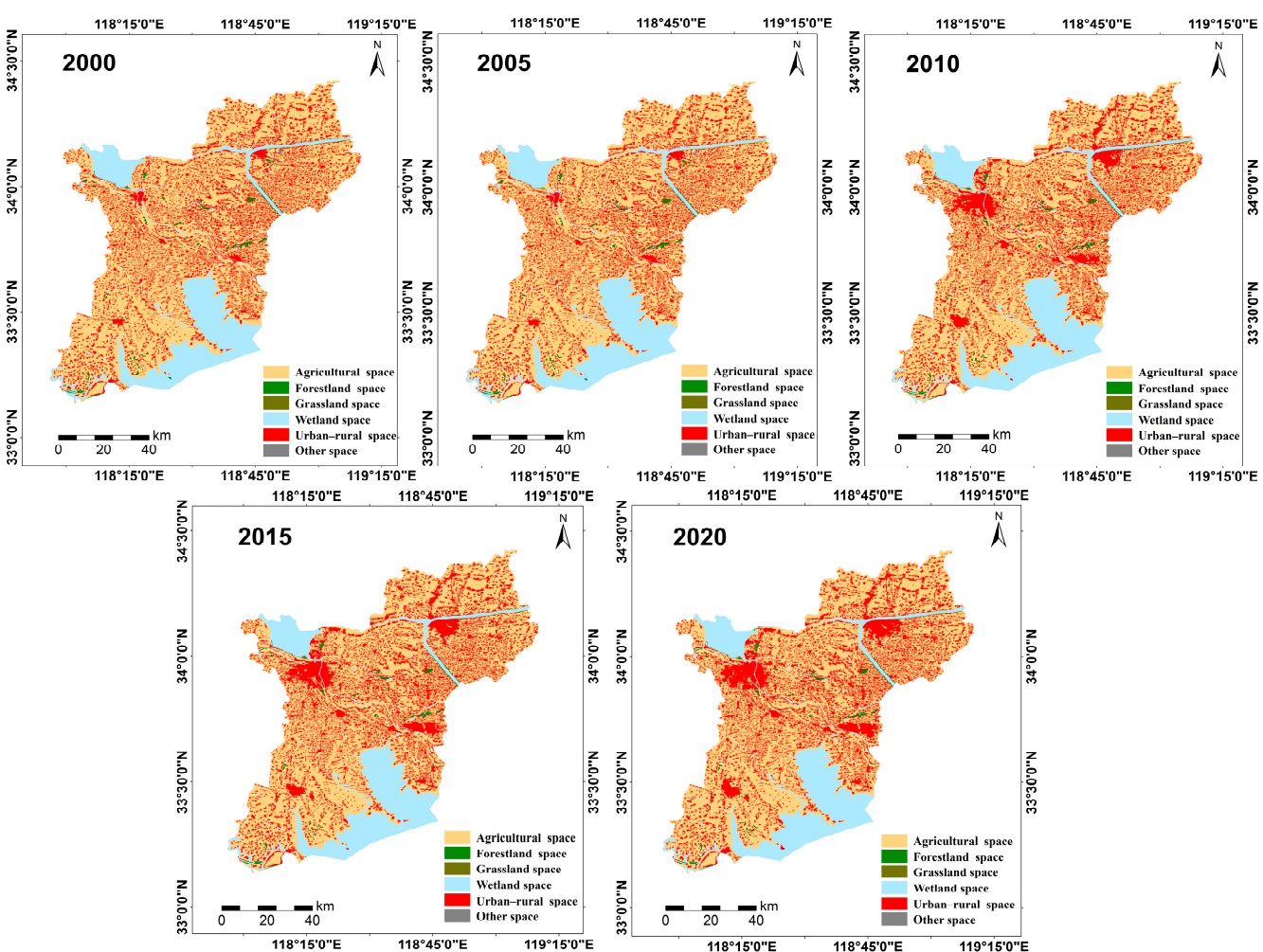

**Figure 3.** Territorial space utilization status of Suqian City from 2000 to 2020.

From 2000 to 2020, there were certain changes in the distribution of territorial space (Figure 4). In 2010, there was an obvious expansion of urban–rural areas outward, mainly because Suqian City took on industrial development and urban-driven strategy as its core development strategy, and the central built-up area continuously expanded. Construction land gradually developed from a scattered small group system to a concentrated group system. At the plain where two rivers intersected, it is clear that urban–rural space continued to expand, and the density of urban–rural area throughout the region significantly increased. The spatial pattern of forestland space changed from centralized distribution

in the northwest and southwest hillocks in 2000 to centralized distribution in the northwest and sporadic distribution in the southwest and central regions. The distribution of grassland space, which was scattered along the lakeshore in the northwest and southwest in 2000, expanded to the outskirts of some towns in the central and southeast regions far away from the lakeshore. Agricultural space did not change much, and it was still widely distributed in the central and eastern regions, but it could be clearly seen that the agricultural space around the urban–rural space was heavily eroded. This was mainly due to the rapid urbanization policy guidance, the center of Suqian City framework continuing to expand, and infrastructure construction making rapid progress. The transportation network, industrial plants, residential houses, etc. squeezed the agricultural space.

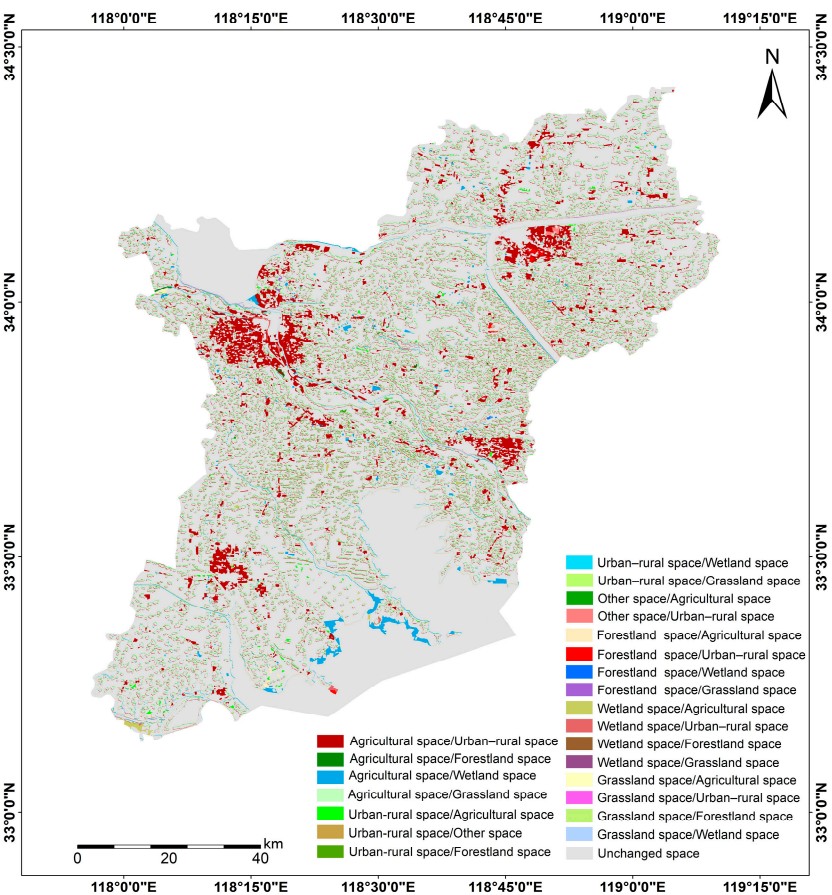

**Figure 4.** The territorial space transfer situation in Suqian from 2000 to 2020.

## 3.2. Analysis of Spatiotemporal Differentiation of Carbon Storage in Suqian City

### 3.2.1. The Carbon Storage Structure in Suqian

The carbon storage in Suqian City is highest in the agricultural space, due to its large area of cultivated land and various types of crops, such as rice, corn, wheat, beans and vegetables, which have high aboveground and underground biomass and thus have a high carbon storage capacity. The second highest is the urban–rural development space, due to its high green coverage rate. The third is the aquatic and wetland ecological space, as Suqian City has two large freshwater lakes, Luoma Lake and Hongze Lake, with wetlands and rivers throughout the region. Water bodies, lake bodies and aquatic vegetation all have certain carbon storage capabilities, but their carbon density values are relatively lower than those in the agricultural production space. Forest and grassland ecological spaces have higher carbon density values than other spaces, but their carbon storage capacity is relatively low due to their smaller coverage areas.

Regarding changes in different periods, the carbon storage in Suqian City slowly decreased from 2000 to 2005, then rapidly decreased from 2005 to 2010, and recovered

somewhat from 2010 to 2015 but then rapidly declined from 2015 to 2020. In 2020, the carbon storage capacity of Suqian City was 78.15 Tg (1 × 106 ton). From 2000 to 2020, the city's carbon storage showed a continuous downward trend, with a total loss of 1.23 Tg of carbon storage due to the "landscape" transformation, with the greatest decrease from 2005 to 2010, reaching 0.84 Tg. Overall, from 2000 to 2020, the agricultural space in Suqian City continued to decrease, with a total decrease of 4.45 Tg.

Regarding changes in different counties (Figure 5a), the overall carbon storage levels in the different counties of Suqian City demonstrated a slight yet stable downward trend. Sihong County had the largest carbon storage value, with values of 24 Tg or more maintained at all stages, followed by Shuyang County. Carbon storage levels in Suyu District and Sucheng District were relatively low. The carbon storage levels of the five counties showed a tendency toward decline from 2005 to 2010, which slowed in the other stages.

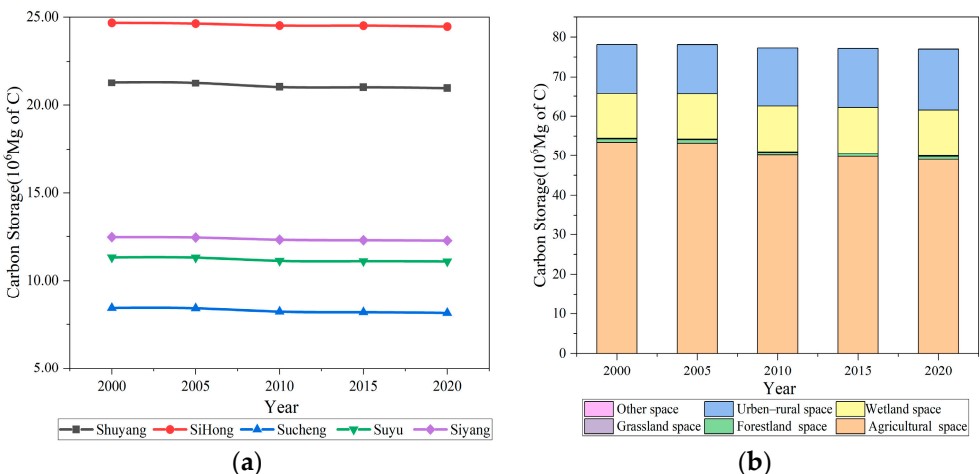

**Figure 5.** (**a**) Carbon storage changes in different districts and counties. (**b**) Carbon storage changes in different territorial spaces.

Regarding changes in different types of spaces (Figure 5b), the change in carbon storage and storage in agricultural space showed was highly consistent with changes in the area of agricultural space over time. Carbon storage in forestland space showed a downward trend, with the rate of decline showing an initial increase followed by a decrease from 2010 to 2020, gradually becoming stable. Carbon storage in the grassland space showed a fluctuating upward trend, with a slight decrease in carbon storage from 2000 to 2010. Carbon storage in the aquatic and wetland space steadily increased, with a slower increase in carbon storage from 2010 to 2015, while the other three stages maintained a carbon storage rate of approximately 0.1 Tg. The increase in carbon storage in urban–rural space continued, with a total increase of 3.01 Tg. The rate of increase initially accelerated and then tended to stabilize.

3.2.2. The Carbon Storage Distribution Pattern in Suqian City from 2000 to 2020

Five periods of territorial space data in Suqian City and the construction of a national spatial carbon density table were input into the carbon storage module of the InVEST model, to obtain a spatial distribution map of carbon storage in Suqian City (Figure 6). Overall, the moderate level of carbon storage, approximately 1 ton, was almost ubiquitous throughout Suqian City. Due to its large distribution range, the changes during the entire period from 2000 to 2020 were not very obvious. The greatest carbon storage, approximately 1.56 tons, was mainly distributed in the northwest and southwest forest areas. The distribution of the blocks in the northwest forest area in 2000 became sporadic, indicating that the carbon storage capacity of the northwest forest area weakened year by year. Between 2000 and 2020, there was a significant decline in medium-level carbon storage, which was closely related to the continuous compression of agricultural space. Suqian City has high water

resource endowment and a wide range of wetland areas. The municipal government continued to carry out the restoration and protection of aquatic and wetland ecological environments, which led to a steady increase in the carbon storage value of waterbodies and wetlands in the study area. Since 2010, many low-level carbon storage areas have been scattered in the study area, mainly concentrated near the central urban area of the district and county, and have continued to expand outward, encroaching on the median and high-value areas. These areas should enhance their awareness of ecological protection, promote the improvement in carbon storage value driven by changes in national ecosystem cover types, and enhance the level of regional carbon sinks.

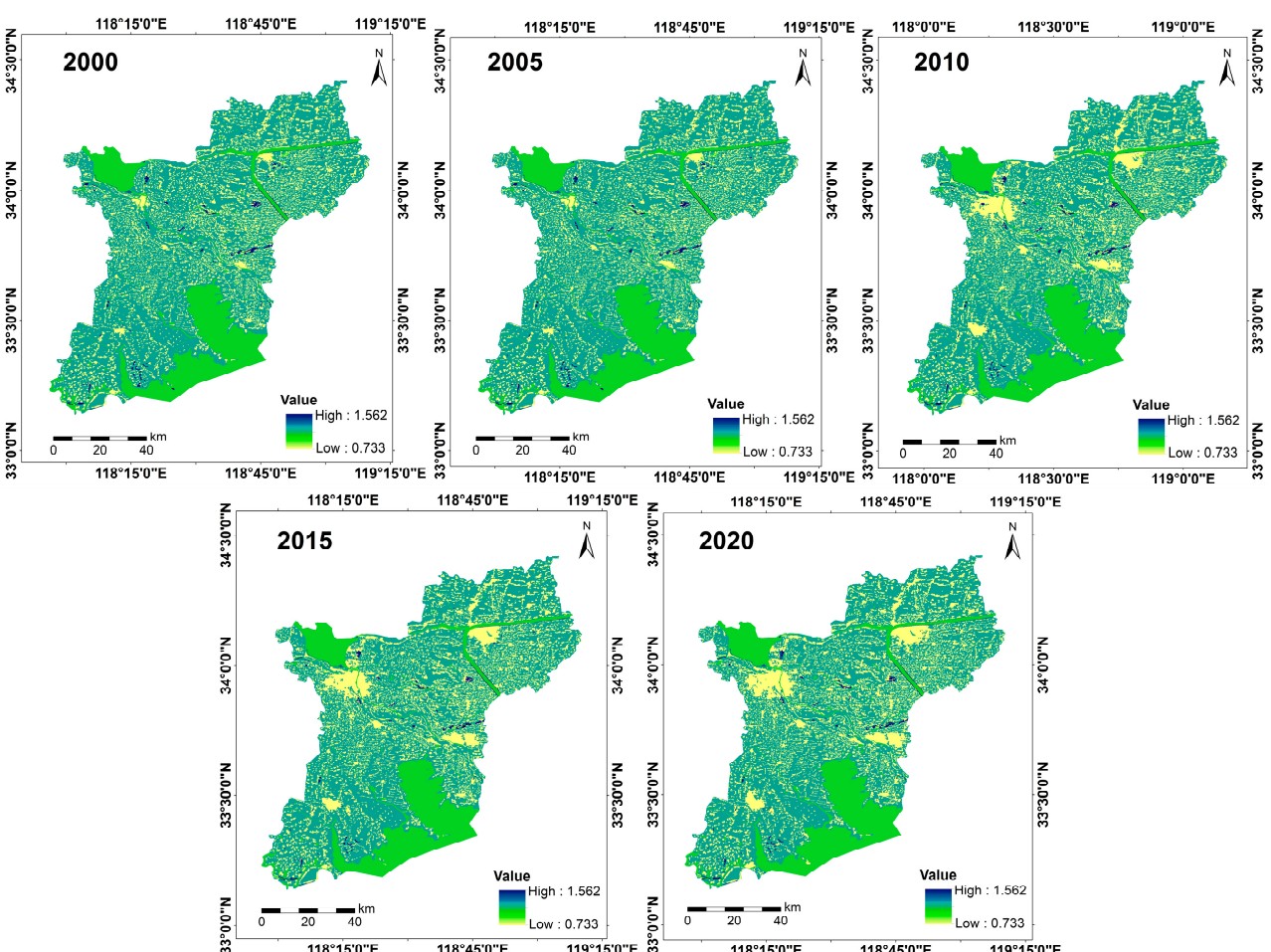

**Figure 6.** The spatial distribution of carbon storage in Suqian from 2000–2020.

The carbon pool refers to the part of the Earth's system that stores carbon in the carbon cycle. Based on the contribution of the carbon pool to the changes in $CO_2$ concentration in the global atmosphere, the carbon pool can be divided into two types, carbon sources and carbon sinks. According to the difference of carbon storage values in the two periods, the study area is divided into carbon storage areas and carbon source areas.

Using ArcGIS 10.5 and the carbon storage grayscale map of Suqian City from 2000 to 2020, the spatial distribution map of carbon storage for different research periods was obtained and the study area was divided into carbon sink areas, carbon source areas and carbon balance areas. Since carbon storage varied in different years, the division criteria were not completely consistent. The critical value was automatically generated by ArcGIS, based on the total carbon storage. The red areas with large absolute negative values represent carbon source areas, the white areas near 0 values represent carbon balance areas, and the green areas with large absolute positive values represent carbon sink areas. Since various ecosystems exist in Suqian City and the spatial partitioning of land cover is not very

clear, the carbon source and carbon sink areas are also relatively scattered and overlap with each other. Suqian's carbon sources and sinks for 2000–2020 ranged from −0.83–0.83 tons, and the numerical changes in carbon sources and carbon sinks were divided into the following five categories: significant decrease (−0.83−−0.415), slight decrease (−0.415–0), basically unchanged (0), slight increase (0–0.415) and obvious increase (0.415–0.83), as shown in Figure 7. The area and proportion of grids in different numerical ranges were calculated, as shown in Table 8.

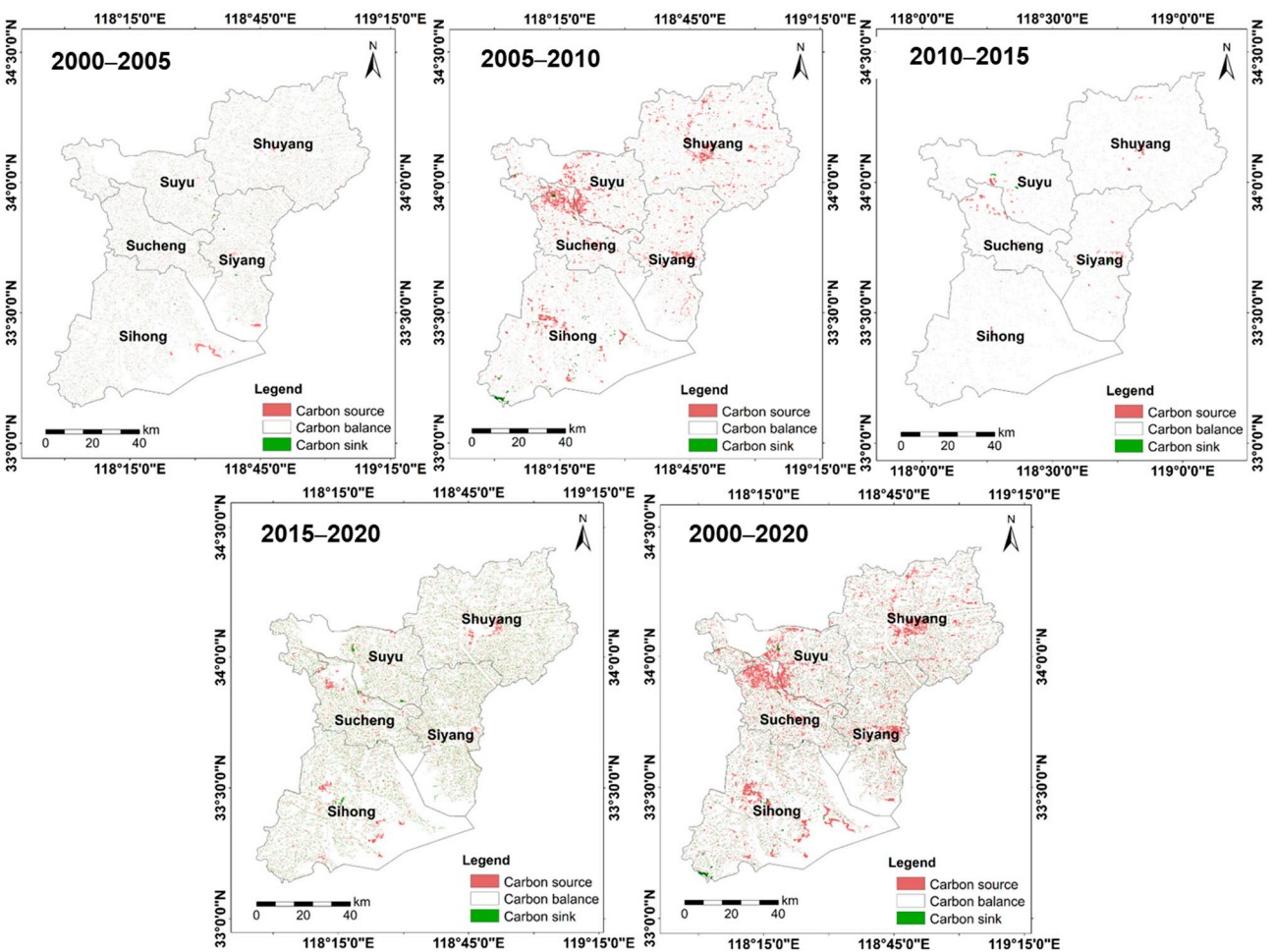

**Figure 7.** The carbon storage distribution in Suqian City from 2000 to 2020.

**Table 8.** Change area (km²) and proportion (%) of carbon storage in Suqian from 2000 to 2020.

| Classification | Scope of Change (t) | Area (km²) | Proportion (%) |
|---|---|---|---|
| Significant decrease | −1.245 | 17.6299 | 0.21% |
| Slight decrease | −0.415 | 786.1857 | 9.22% |
| No change | 0 | 7376.778 | 86.56% |
| Slight increase | 0–0.415 | 330.5294 | 3.88% |

Between 2000 and 2020, Suqian's carbon source areas were concentrated in northern Sucheng District, urban centers in Shuyang County, northern Sihong County, eastern Siyang County and along the Hongze Lake Coast, and some carbon source areas were scattered around the residential areas in the city. Carbon storage was reduced by 17.63 km², or 0.21% of the total area, mainly in western Sucheng District, Central Shuyang County, and the southwest coast of Hongze Lake, and the slight reduction in carbon storage accounted for 9.22% of the total area. Carbon sink areas were mainly distributed in the southwest and

northwest hilly areas, with some scattered around the towns and riverbanks. The carbon sink area increased significantly, accounting for 0.13% of the total area. The area of carbon storage increased slightly, accounting for 330.53 km$^2$ or 3.88% of the total area. The area of carbon storage was mainly distributed in the southeastern boundary of Sihong County and Santaishan National Forest Park. The carbon balance area was still widely distributed in the central and eastern parts of Suqian, with an area of 7376.78 km$^2$, accounting for 86.56% of the total area.

### 3.3. Analysis of the Carbon Storage Effect of National Land Transition in Suqian City

3.3.1. The Relationship between National Land Transition and Change in Carbon Storage

This chapter details the changes in carbon storage caused by changes in different land covers using the territorial space transfer matrix and carbon storage table from 2000 to 2020 in Suqian City. Notably, for each type of space, the calculated change in carbon storage only considers the vegetation and soil carbon storage as each type of space changes and does not consider the increase or decrease in carbon storage before and after the conversion of space. The changes in vegetation carbon storage and soil carbon storage according to changes in landscape from 2000 to 2020 can be seen in Table 9.

**Table 9.** Changes in carbon density and storage caused by territorial spatial transformation.

| Territorial Space Conversion | Vegetation Carbon Density (t/hm$^2$) | Soil Carbon Density (t/hm$^2$) | Vegetation Carbon Storage (10$^3$ t) | Soil Carbon Storage (10$^3$ t) | Total (10$^3$ t) |
|---|---|---|---|---|---|
| Agricultural space–Forestland | 22.9 | 34 | 14.73 | 21.92 | 36.66 |
| Agricultural space–Grassland | 4.9 | 6.7 | 1.64 | 2.27 | 3.91 |
| Agricultural space–Wetland | −5.5 | **−11.7** | −45.10 | −95.37 | −140.47 |
| Agricultural space–Urban–rural space | −6.4 | −19.7 | −435.06 | −1346.73 | −1781.78 |
| Subtotal | - | - | −463.78 | −1417.90 | −1881.68 |
| Forestland–Agricultural space | −36.8 | −34 | −27.06 | −25.00 | −52.06 |
| Forestland–Grassland | −30.9 | −27.3 | −2.79 | −2.46 | −5.25 |
| Forestland–Wetland | −48.3 | −45.7 | −10.97 | −10.38 | −21.35 |
| Forestland–Urban–rural space | −57.5 | −53.7 | −40.89 | −38.20 | −79.09 |
| Subtotal | - | - | −81.71 | −76.04 | −157.75 |
| Grassland–Agricultural space | −4.9 | −6.7 | −0.07 | −0.10 | −0.17 |
| Grassland–Forestland | 18 | 27.3 | 0.00 | 0.01 | 0.01 |
| Grassland–Wetland | −10.4 | −18.4 | −3.54 | −6.27 | −9.81 |
| Grassland–Urban–rural space | −11.2 | −26.4 | −0.15 | −0.35 | −0.50 |
| Subtotal | - | - | −3.76 | −6.72 | −10.48 |
| Wetland–Agricultural space | 5.5 | 11.7 | 18.35 | 38.80 | 57.15 |
| Wetland–Forestland | 28.4 | 45.7 | 6.55 | 10.55 | 17.10 |
| Wetland–Grassland | 10.4 | 18.4 | 2.20 | 3.91 | 6.11 |
| Wetland–Urban–rural space | −0.8 | −8 | −1.14 | −10.87 | −12.01 |
| Subtotal | - | - | 25.97 | 42.38 | 68.35 |
| Urban–rural space–Agricultural space | 6.4 | 19.7 | 181.08 | 560.54 | 741.62 |
| Urban–rural space–Forestland | 29.2 | 53.7 | 1.15 | 2.70 | 3.85 |
| Urban–rural space–Grassland | 11.2 | 26.4 | 0.50 | 4.73 | 5.22 |
| Urban–rural space–Wetland | 0.8 | 8 | 0.50 | 4.73 | 5.22 |
| Urban–rural space–Other space | −0.9 | 1.6 | −0.01 | 0.01 | 0.00 |
| Subtotal | - | - | 183.21 | 572.71 | 755.92 |
| Other space–Agricultural space | 7.2 | 18.1 | 0.07 | 0.17 | 0.23 |
| Other space–Urban–rural space | 0.9 | −1.6 | 0.24 | −0.44 | −0.20 |
| Subtotal | - | 2.9 | 0.30 | −0.27 | 0.03 |
| Total | - | - | −339.77 | −885.84 | −1225.61 |

Overall, without considering the carbon storage change after territorial space conversion, changing spatial patterns in space use decreased the organic carbon storage in Suqian

city by 1225.61 thousand tons from 2000 to 2020, including a decrease of 885.84 thousand tons in soil organic carbon storage and 297.61 thousand tons in vegetation carbon storage. The final analysis of the types of ecosystems shows that converted agricultural space, forestland space and grassland space resulted in a decrease in organic carbon storage by 1881.68 thousand tons, 157.75 thousand tons and 10.48 thousand tons, respectively, while converted wetland space, urban–rural space, and other space resulted in an increase in organic carbon storage by 68.35 thousand tons, 755.92 thousand tons and 0.03 thousand tons, respectively. Among them, the conversion of agricultural space to wetland space, urban–rural space and other space, the conversion of grassland space to urban–rural space and other space, the conversion of wetland space to other space and urban–rural space and finally the conversion of urban–rural space to other lands all led to a decrease in both soil and vegetation organic carbon storage, which was not conducive to the formation of carbon sinks. Among them, it should be noted that the encroachment of agricultural space by urban–rural space led to a decrease of 1781.78 thousand tons of organic carbon storage, accounting for 86.92% of the total reduction. The transformation from agricultural space to grassland space led to a decrease in vegetation organic carbon storage, but it was beneficial for increasing soil carbon storage.

3.3.2. The Contribution of Territorial Spatial Transformation to Changes Carbon Storage

According to the formula-based calculation method of carbon storage, the contribution rate of carbon storage due to each type of space transformation can be obtained (Figure 8). The mutual transformation between agricultural production space and urban–rural construction space dominated the changes in carbon storage in Suqian City. Among the space use transformations leading to an increase in carbon storage, urban–rural construction space converted to agricultural production space contributed most, followed by the transformation of wetland space into agricultural space, and the third was the transformation of agricultural space into forestland space. These three transformations account for a total proportion of 95.25%. The transformation of agricultural space into other types of space became the key factor for increasing carbon storage. The spatial transformation from agricultural space to urban–rural space was the main contributing factor to the reduction in carbon storage, followed by the transformation of agricultural space into wetland space and then the transformation of forestland space into agricultural space and urban–rural space. The transfer of agricultural space and forestland space were the main factors contributing to the decrease in carbon storage, with a contribution rate of 97.66%.

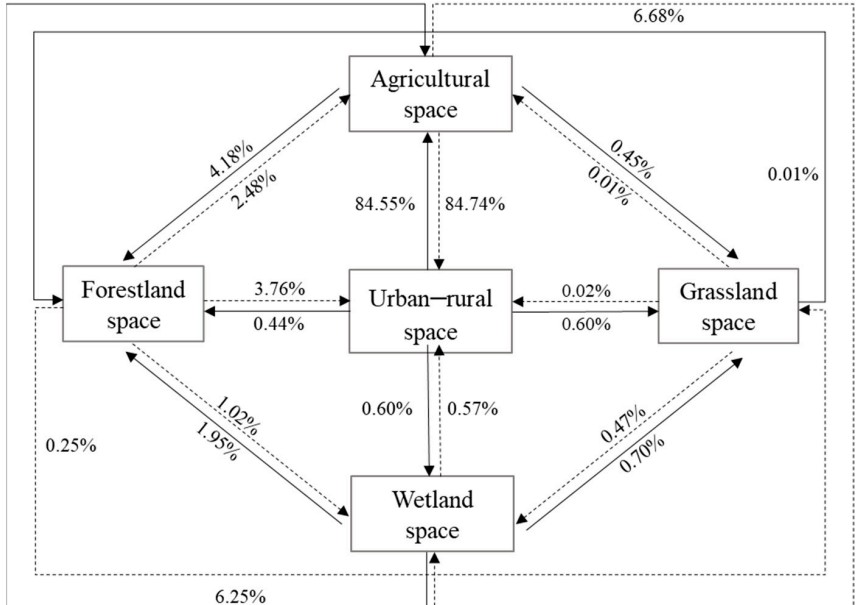

**Figure 8.** Carbon storage contribution of national land transformation.

## 4. Discussion

### 4.1. Territorial Space Conversion and Its Carbon Storage Effect

The transformation of territorial space is essentially the evolution of spatial patterns due to meeting human demands along with the changes and developments of society. This transformation is directly affected by changes in the human–land relationship, and its comprehensive effects have a feedback effect on human society. In the analysis of space transfer, the encroachment of urban–rural space on agricultural space is the main pattern in Suqian City. Currently, this is a common feature of China's rapid urbanization [39,40]. The expansion of urban–rural space in Suyu District and Sucheng District is more obvious than in the other three counties. This is because these two places are the economic engines of the city, with a high level of urbanization, complete infrastructure construction and a developed industrial and commercial sector. The most drastic period of territorial spatial transformation was from 2005 to 2010. Urban expansion may reduce carbon storage by reducing vegetation cover and function, ultimately leading to a decrease in carbon storage through the occupation of surrounding farmland, forests and other ecological environments [41]. However, this expansion is different from that of some developed countries, such as the United States [42], Australia [43] and Europe [44,45], as these countries have advanced further in the urbanization process.

According to the research, Suqian City has relatively high soil carbon density in urban–rural space, which is consistent with previous studies [46,47]. This is mainly due to the diverse sources of soil carbon in urban areas, including leaves, branches, weeds, large amounts of household waste and organic waste generated by urban industries [48]. Although central cities with residential areas as their main land type are considered important carbon sources, the vegetation in residential areas, including urban green spaces, parks and small urban forests, can play an important role in carbon storage. The continuous expansion of cities will undoubtedly threaten ecological security, but increasing the green space ratio in urban areas seems to be an effective way to offset carbon loss. Studies have shown that forestland space exhibits the highest comprehensive carbon density, as it can enhance carbon storage in soil by maintaining a high level of biomass and creating more residual vegetation. Therefore, planting high-biomass green vegetation in urban areas is crucial for carbon accumulation. The conversion of agricultural space is one of the main reasons for the significant decrease in carbon storage, which is consistent with previous studies. On the one hand, the vegetation biomass of agricultural land is higher than that of all other land use types, except forestland and grassland. On the other hand, factors affecting soil carbon storage, such as returning crop residues and straw, long-term use of organic fertilizers and the flooding of paddy fields, also contribute to enhancing carbon storage to some extent [49,50]. Wetland space not only have great potential for carbon storage but also have functions in water conservation, flood control, erosion control, fuel and food production and biodiversity enhancement [51]. Although the wetland area in the study area is relatively lacking in vegetation, it still shows a relatively high soil carbon density. This is mainly because the water transport mechanism collects organic carbon from other areas and accumulates it in the sediment at the bottom of the water [52]. Therefore, it is necessary to consider wetlands in planning and optimization.

### 4.2. Carbon Storage Optimization Strategy under Low-Carbon Targets

As a complex system of "economy–society–ecology", the territorial space refers not only to land space or land use itself but also to regional units with specific structures and functions that encompass natural processes and human activities [53]. Therefore, optimizing carbon storage in the territorial space requires a systemic perspective, considering factors such as spatial functions, scale, structure and layout. It involves exploring land management methods that contribute to reducing the impact of carbon emissions transfer and promoting regional low-carbon development. Additionally, it involves adapting to different patterns of territorial spatial transformation and exploring differentiated territorial

layout models and development patterns to achieve the dual goals of carbon storage and emission reduction (Figure 9).

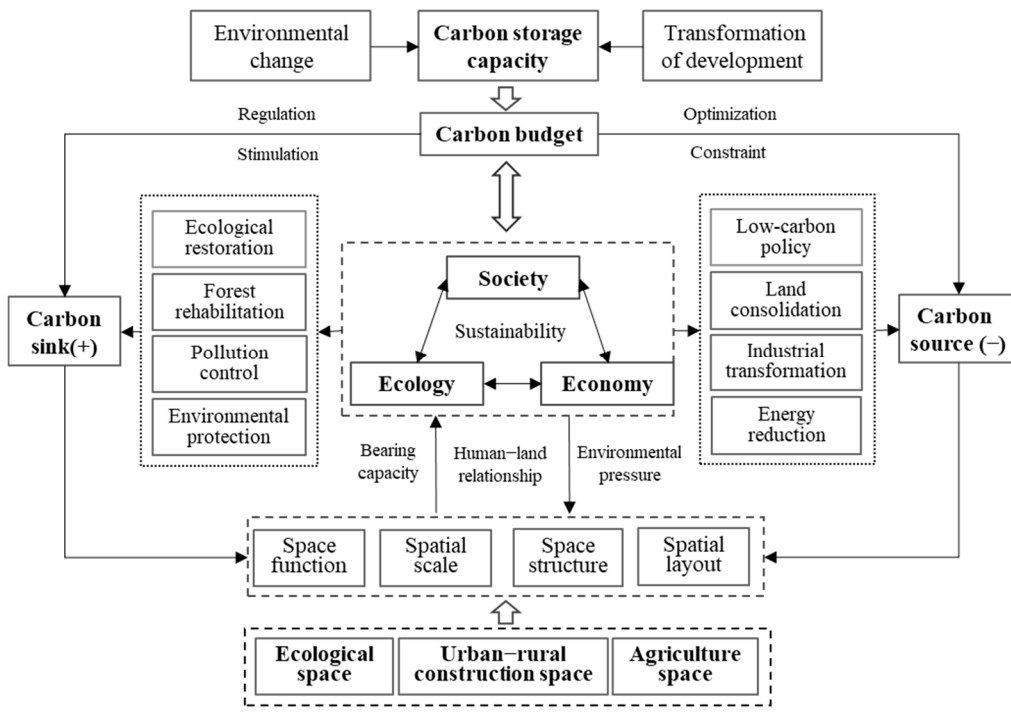

**Figure 9.** Mechanism for regulation of carbon storage capacity in territorial space planning.

From the perspective of a complex system, increasing the capacity for carbon storage in territorial space needs to be achieved by regulating the functions, scale, structure and layout of territorial space. Territorial space function regulation is based on identifying the carbon source/sink functions of land and evaluating the rationality, feasibility and possibility of land transitioning toward carbon sink functions or decreasing carbon source/sink functions. Examples include land reclamation and afforestation. Territorial space scale regulation involves restricting the expansion of land with carbon source functions and incentivizing the expansion of land with carbon sink functions, for example by ensuring the strict control of construction land and promoting desert greening. Territorial space structure regulation aims to control the proportion of land with carbon source/sink functions to promote local carbon absorption, such as increasing the ratio of affiliated green spaces in residential areas. Territorial space layout regulation focuses on controlling the location and use type to promote carbon emission reduction and storage, such as improving clean energy utilization through the local distribution of industrial and clean energy lands, reducing spatial and temporal frictions through the local distribution of residential and public facility lands and enhancing carbon sink capacity through the layout of green corridors and strips within green spaces. For other hierarchical spaces, such as ecological, agricultural and urban spaces, the capacity for carbon storage and regulation should address key issues in scaling up spatial dimensions, such as controlling the proportions of ecological, agricultural, and urban spaces within a region. Regulation also involves controlling the location, form and routing of urban and agricultural spaces in relation to infrastructure land.

### 4.3. Uncertainty and Limitations

This study has some limitations that need to be addressed in future research. First of all, the LULC data used in the study are not accurate enough, which would impair the assessment of carbon storage. The overall accuracy of the land cover map used in this study is approximately 79.8–86.1%. Although the accuracy of the map meets the scientific criteria for regional scale studies, it still affects the accuracy of the results of

land classification identification and extrapolation of policy implications. The inclusion of village settlements as built-up areas may lead to an overestimation of the built-up area and thus an underestimation of carbon storage. Second, it is a widely used method to obtain the carbon density parameters of different land space from a priori data for the InVEST model [54–56]. But most studies assume that carbon density remains constant over time. When authors obtain carbon density data for different territorial spaces from the published literature, some uncertainty may be introduced due to inconsistent research periods. Calculating the average carbon density of vegetation and soil in different regions by overlaying land use maps and climate zone maps may be an effective method to improve estimation accuracy [57]. Additionally, the InVEST model may not be able to take into account some specific situations in the study area. The carbon storage module acquiesces that the transformation of different territorial space is the only reason for carbon storage change, and neglects the natural recovery and succession process of regional territorial space [58,59]. Some studies use the InVEST model to estimate carbon storage while also calculating the economic value of carbon storage [60]. However, this study mainly explores the direct impact of territorial spatial transformation on carbon storage, without taking into account economic value. Furthermore, this study assumes an immediate change in carbon storage after LULC conversion, while the inevitable transition to a new steady state may take decades [61]. In future research, further exploration is needed to assess the spatiotemporal relationship between multiple ecosystem services in the study area and natural factors as well as environmental policies.

**5. Conclusions**

Based on the theory of territorial spatial transformation and space use planning, this study analyzed the characteristics of territorial space evolution and its impact on carbon storage in Suqian City using methods such as dynamic degree, transition matrix, and the InVEST model. The following conclusions were drawn:

(1) The conversion between agricultural space and urban–rural space in Suqian City was the largest from 2000 to 2020, with urban–rural space occupying 684.42 km$^2$ of agricultural space. The area of wetland space showed a stable growth trend, while forestland space exhibited a continuous decline.

(2) Carbon storage in Suqian City showed a gradual decrease, with a total reduction of $1.23 \times 10^6$ tons over 20 years and a decrease of 1.46% compared to the initial value. Forest spaces had significantly higher carbon density than other spaces. The conversion from agricultural space to urban–rural construction space was the dominant factor leading to an increase in carbon storage. Conversely, the conversion from agricultural production space to urban–rural construction space and wetland space was the main reason for the significant reduction in carbon storage.

(3) The mechanisms and specific strategies for optimizing space use planning under low-carbon goals were determined based on the transformation of the territorial space and carbon storage change in the study area. To enhance carbon storage capacity, optimizing the planning of territorial space requires both an overall regulation of spatial functions, scale, structure and layout and the differentiated management and regulation of specific spaces. This will effectively enhance the regional carbon storage capacity.

**Author Contributions:** W.H.: conceptualization, formal analysis, methodology and writing—original draft. L.G.: data curation, investigation and writing—review and editing. T.Z.: conceptualization, funding acquisition and supervision. T.C.: investigation, visualization and writing—original draft. L.C.: validation, visualization and writing—original draft. L.L.: visualization and writing—original draft. X.Z.: investigation and writing—original draft. All authors have read and agreed to the published version of the manuscript.

**Funding:** This research was supported by the Research Project on Enhancing Path and Regulation Mechanism for Carbon Storage Capacity of Territorial Spatial Ecosystem of Suqian City (Grant No.: 2022170002).

**Data Availability Statement:** The original contributions presented in the study are included in the article, further inquiries can be directed to the corresponding author.

**Acknowledgments:** The authors are grateful to the editor and reviewers for their valuable comments and suggestions.

**Conflicts of Interest:** The authors declare no conflicts of interest.

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
