# Peer review of "The Impact of Territorial Spatial Transformation on Carbon Storage: A Case Study of Suqian, East China"

_land, doi:10.3390/land13030348_

Round 1
Reviewer 1 Report
Comments and Suggestions for Authors
Useful research, thoroughly conducted.
Author Response
Dear Reviewer,
On behalf of my co-authors, we are very grateful to you for giving us an opportunity to revise our manuscript. We do appreciate your positive and constructive comments and suggestions on our manuscript entitled “The Impact of Territorial Spatial Transformation on Carbon Storage: A Case Study of Suqian, East China” (Manuscript ID: land-2858146).We will continue our research and try to fill the gaps.
Thanks again to the hard work of you !
Reviewer 2 Report
Comments and Suggestions for Authors
The article is related to the extremely relevant issue of greenhouse gas emissions. The main goal was to evaluate the impact of space transformation on carbon storage. To achieve this, the authors conducted research in Suqian city.
The article begins with an extensive literature review. Some concerns arise regarding the reliance of the review almost exclusively on Chinese authors' works (with one foreign reference from Minnesota). While the study focuses on China, it might be worthwhile to broaden the perspective on the topic. The article clearly and comprehensively presents the research methodology. The results are presented step by step in a way that leaves no room for doubt. The strength of the article lies in the graphics and maps, which make it easy to understand the studied area. The objectives set in the introduction have been achieved.
A few minor suggestions:
- Clearly articulate the main goal of the research more strongly. It is mentioned in the article but not very precisely.
- As is typical in such studies, the coefficients derived from other research (Table 1) are a matter of discussion. They are heavily averaged. In the future, it might be valuable to consider a more detailed approach that takes into account the actual carbon storage capabilities, as the authors noted in the "Uncertainty and Limitations" section.
- Provide a more detailed description of the InVEST module.
- Similar to the introduction, there is a slight lack of reference to research results conducted outside of China. They are summarized in one sentence (l.533). It seems worthwhile to expand on this point.
- Consider recalculating certain results per hectare (only to be considered).
Overall, the article is of a very high scholarly standard.
Author Response
Thank you very much for your valuable suggestion.Please see the attachment.

Reviewer 3 Report
Comments and Suggestions for Authors
This is a nice paper I would like to see in print, and soon.
I. The biggest issue is that the paper needs a more thorough proofreading as line 275 has “change” in stead of “changed” and line 283 had “nation” in place of “national” as typical examples of the small mistakes littered through the text that make it look rushed.
II. The paper makes excessive use of the first person. This makes it sound more like a conference presentation before an audience. I suggest changing first person references (we) to third person to make the language sound more formal and authoritative.
III. Figure 2 is a little confusing as the lines heavily overlap and the colors are not as dramatically different as they could be to make sure what the differences are. Maybe this could be presented as two figures with each one larger to be easier to read or maybe as a table or maybe some other sort of figure.
IV. Tables 2-6 are clear, but fairly dense. Could the outliers be made bold or italicized to draw attention to them?
V. Figures 3 and 6 are not terribly clear, with colors that do not stand out as well ti indicate the major changes. This is particularly the case for the Urban-Rural expansion, which would be clearer in something like black. The pale wetlands color has the clearest contrast but it has barely any change at all. The most dramatic changes should be the most visible. Another strategy would be to have larger figures that are easier to see in detail.
VI. Table 7 uses “reduce” instead of “reduction.”
VII. There are two Figure 8s.
VIII. I had other comments but they were mostly answered by rereading the text the second and third times.
Comments on the Quality of English LanguageThere are numerous small errors in the text that need to be corrected with a more thorough proofreading. Above are given a few examples.
Author Response

(The authors gave the same response as above.)

Reviewer 4 Report
Comments and Suggestions for Authors
The paper effectively addresses the issue of urbanization and land use changes in Suqian City, emphasizing the importance of maintaining carbon storage in terrestrial ecosystems. The paper connects the changes in territorial space to carbon emissions, highlighting the significance of land use planning in mitigating climate change. This connection is crucial, considering the current global emphasis on achieving carbon neutrality.The paper describes the methodology used for the study, including the dynamic degree of territorial space, transition matrix, and the InVEST module for carbon storage estimation. This transparency in the research process enhances the credibility of the study.
The introduction is well-written and provides context. Suggest adding a sentence briefly summarizing the specific objectives of the study at the end of the introduction. Consider incorporating recent research or findings related to the impact of land use changes on carbon storage. This could strengthen the background information. Materials and Methods: Clearly define the abbreviations used in the paper. Provide more details on the adjustments made to the carbon density data based on temperature and precipitation factors. Ensure consistency in the use of units across the paper, especially in the carbon density data table and calculations.
Author Response

(The authors gave the same response as above.)
